# Physical Therapists’ Ethical and Moral Sensitivity: A STROBE-Compliant Cross-Sectional Study with a Special Focus on Gender Differences

**DOI:** 10.3390/healthcare11030333

**Published:** 2023-01-23

**Authors:** Noemí Moreno-Segura, Laura Fuentes-Aparicio, Sergio Fajardo, Felipe Querol-Giner, Hady Atef, Amalia Sillero-Sillero, Elena Marques-Sule

**Affiliations:** 1Department of Physiotherapy, University of Valencia, 46010 Valencia, Spain; 2Physiotherapy in Motion, Multispeciality Research Group (PTinMOTION), Department of Physiotherapy, University of Valencia, 46010 Valencia, Spain; 3Mersey Care NHS Trust, Harltey Hospital, Southport, Merseyside PR8 6PL, UK; 4Department of Physical Therapy for Cardiovascular, Respiratory Disorder and Geriatrics, Faculty of Physical Therapy, Cairo University, Giza 11485, Egypt; 5ESIMar (Mar Nursing School), Parc de Salut Mar, Universitat Pompeu Fabra affiliated, 08002 Barcelona, Spain; 6SDEHd (Social Determinants and Health Education Research Group), IMIM (Hospital del Mar Medical Research Institute), 08002 Barcelona, Spain

**Keywords:** physical therapy, professional practice, ethical sensitivity, moral sensitivity, ethics

## Abstract

(1) Background: Healthcare professionals´ clinical practice, their care of patients and the clinical decision-making process may be influenced by ethical and moral sensitivity. However, such outcomes have been scarcely studied in physical therapists. This study aimed to explore ethical sensitivity and moral sensitivity in practicing physical therapists, and to compare both variables by gender. (2) Methods: Cross-sectional study. 75 physical therapists (58.7% women; average age = 34.56 (8.68) years) were asked to fill in questionnaires measuring ethical sensitivity (Ethical Sensitivity Scale Questionnaire) and moral sensitivity (Revised-Moral Sensitivity Questionnaire). (3) Results: The sample showed high ethical sensitivity (116.14 ± 15.87 over 140) and high moral sensitivity (40.58 ± 5.36 over 54). When comparing by gender, women reported significantly higher ethical sensitivity than men (*p* = 0.043), as well as higher scores in the following dimensions: Caring by connecting with others (*p* = 0.012) and Working with interpersonal and group differences (*p* = 0.028). However, no differences were found in moral sensitivity (*p* = 0.243). (4) Conclusion: Physical therapists showed high levels of ethical and moral sensitivity, whilst women reported higher ethical sensitivity than men. Understanding physical therapists´ ethical and moral sensitivity is essential to design and implement integrated education programs directed to improve the quality of care of patients in their daily clinical practice.

## 1. Introduction

Healthcare professionals´ clinical practice and quality of care of patients may be influenced by ethical aspects, such as ethical and moral sensitivity, that could affect their daily clinical practice. In this sense, ethics currently has an important role in healthcare professions and especially in the physical therapy field [1,2,3,4,5,6]. It is expected that healthcare professionals have extensive intellectual and practical education to commit themselves to the welfare of those in their care [7]. However, research related to ethical aspects in practicing physical therapists, including examples such as ethical dilemmas that may arise in clinical practice, is scarce [8], while ethics-related outcomes have been extensively studied in several healthcare professions such as nursing or medicine. This may be due to the fact that physical therapy is a recent profession compared to other healthcare professions [9].

Healthcare professions care for patients, as well as giving humanized attention to the patients. Humanized attention means ethical care, which enables practitioners to understand the patients in a global way including cognitive, affective and social skills [10]. Moreover, physical therapy is characterized by physical contact with patients during the treatment sessions. Since most of the main therapeutic techniques require direct contact, this fact may lead to a close relationship between the physical therapist and the patient, even closer than in other healthcare professions [11,12]. Therefore, the relationship between physical therapists and patients may lead to ethical problems, especially within the workplace [13,14]. To achieve optimal results, all of these problems should be solved, and physical therapists must have a global vision of the patient considering the clinical context, their beliefs and values, as well as the active role of the patient [11,12]. In this regard, having an open, critical, tolerant and respectful attitude towards those who express different opinions or beliefs is essential to solve such problems in daily clinical practice [11,12,13,15]. Therefore, physical therapists should be aware of the fundamental values and standards of their profession, as well as the main ethical issues that may arise in clinical practice [3,16,17,18]. In this line, two important basic components that compose the morality should be differentiated: ethical sensitivity and moral sensitivity, which have different nuances that should also be differentiated [18,19,20,21]. 

One the one hand, according to Weaver et al. (2008) [10], ethical sensitivity is defined as “the capacity to decide with intelligence and compassion, given the uncertainty in a care situation, drawing as needed on a critical understanding of codes for ethical conduct, clinical experience, academic learning and self-knowledge, with an additional ability to anticipate consequences and the courage to act” [10]. Based on this definition, the central feature of ethical sensitivity is decision-making capability in the uncertainty of professional practice and involves a cognitive capacity, including feelings, knowledge and moral skills, and an interrelational process. Commonly, these situations prompt choices with a significant impact on the well-being of others [17]. Moreover, previous studies determined that there is an important connection between ethical sensitivity and clinical competency; therefore, this concept is more than a reference to ethical attitude [21]. Ethical sensitivity is composed of seven dimensions: reading and expressing emotions, taking the perspectives of others, caring by connecting with others, working with interpersonal and group differences, preventing social bias, generating interpretations and options, and identifying the consequences of actions and options.

On the other hand, Rest (1982) [22] defined moral sensitivity as “the perception that something one might do or is doing can affect the welfare of someone else either directly or indirectly (through a violation general practice or commonly held social standard)”. Furthermore, Lützén et al. (1995 and 1997) [23,24] defined moral sensitivity as “the ability to identify an ethical problem and understand the consequences of decisions made on the patient’s behalf”. It is not only a matter of ‘feeling’ but a personal capacity to ‘sense’ the moral significance of a situation [12]. Moral sensitivity is composed of three dimensions: a sense of moral burden, moral strength, and moral responsibility.

Therefore, it seems that broadening the perspective of ethical sensitivity and moral sensitivity is important, focusing not only on decision-making but also on physical therapist´s daily clinical care. Thus, taking into account that the quality of care may be influenced by the physical therapists’ ethical and moral sensitivity, both are important outcomes to be measured in healthcare professionals, and specifically in physical therapists. The main objective of this study was to explore ethical sensitivity and moral sensitivity in practicing physical therapists, and to compare both variables by gender.

## 2. Materials and Methods

### 2.1. Design and Setting

This was a cross-sectional study. All participants were fully informed about the study’s purpose and procedures and provided written informed consent prior to participating. This study complies with the Strengthening the Reporting of Observational Studies in Epidemiology (STROBE) [25].

### 2.2. Participants

A total of 75 practicing physical therapists were recruited. The inclusion criterion was to be working as a physical therapist, and the exclusion criterion was to be unemployed.

### 2.3. Procedure

The study was conducted from November 2020 to March 2021. All participants completed the questionnaires in one session. First, an investigator trained in the use of the assessment tools contacted all participants and conducted face-to-face interviews in various healthcare settings. In this interview, the content and procedures of the study were fully explained to all participants, the informed consent was provided, and socio-demographic characteristics were collected. Second, the researcher provided self-reported questionnaires to the participants; thus, responses were given individually by participants and privacy was ensured.

### 2.4. Outcomes

All participants provided sociodemographic characteristics and subsequently filled out two validated self-reported questionnaires to assess ethical sensitivity and moral sensitivity. Outcome measures were as follows.

#### 2.4.1. Ethical Sensitivity

Ethical sensitivity was assessed with the Ethical Sensitivity Scale Questionnaire (ESSQ), which is structured in seven different dimensions: (1) reading and expressing emotions, (2) taking the perspectives of others, (3) caring by connecting with others, (4) working with interpersonal and group differences, (5) preventing social bias, (6) generating interpretations and options, and (7) identifying the consequences of actions and options [19,20,26]. The questionnaire is composed of 28 items (4 items per each dimension) measured with a 5-point Likert scale (1 = totally disagree, 5 = totally agree). Scores range from 28 points (lower ethical sensitivity) to 140 points (higher ethical sensitivity) [20]. The reliability of ESSQ is between α = 0.63 (avoid social bias) and α = 0.79 (work interpersonal and group differences). The reliability of this questionnaire is α = 0.81 [20].

#### 2.4.2. Moral Sensitivity

Moral sensitivity was measured with the Revised Moral Sensitivity Questionnaire (RMSQ) [16]. It is a validated questionnaire that includes nine items that represent the three main dimensions of moral sensitivity: (1) sense of moral burden, (2) moral strength, and (3) moral responsibility. Items were measured using a 6-point Likert scale (1 = totally disagree, 6 = totally agree). Scores range from 9 points (lower moral sensitivity) to 54 points (higher moral sensitivity) [16,27]. The questionnaire has been proven to be a valid and reliable instrument (α = 0.83) to assess moral sensitivity [16].

### 2.5. Statistical Analysis

Statistical analysis was conducted using SPSS Version 26.0 (SPSS Inc., Chicago, IL, (USA)). Descriptive results of continuous data were calculated using mean and standard deviation (SD), while nominal data were described using frequencies and percentages. Normality was assessed with the Kolmogorov–Smirnov test. For inferential analysis, *t*-tests between gender groups were used. The significance level was set at *p* < 0.05.

### 2.6. Ethical Considerations

The study protocol was approved by the Institutional Review Board of the University of Valencia, Spain (IE1544051) and all procedures were conducted according to the principles of the Declaration of Helsinki (October 2013, Fortaleza, Brazil) [28].

## 3. Results

A total of 75 physical therapists (average age = 34.56 ± 8.68; 58.70% women) participated in this study. Sociodemographic characteristics of the sample are depicted in Table 1.

### 3.1. Ethical Sensitivity

Table 2 shows the results of the ESSQ. The ESSQ total score of the sample was 116.14 (15.87) points over 140. Regarding the dimensions of the questionnaire, the maximum score was obtained in the dimension “*Taking the perspectives of others*” (4.61 (0.64) points over 5), while the minimum score was obtained in the dimension “*Preventing social bias*” (3.69 (0.63) points over 5). With regard to the items of the questionnaire, the maximum score was obtained in item 8 “*I try to have good contact with all the people I am working with*” (4.8 (0.73) points over 5) and the minimum score was obtained for item 20 “*When I am resolving ethical problems, I try to take a position out of my own social status*” (3.00 (1.13) points over 5).

When comparing ethical sensitivity by gender, significant differences were found in the ESSQ total score; thus, women showed higher ethical sensitivity than men (119.21 (16.26) vs. 111.43 (14.28), respectively, *p* = 0.043). Statistically higher results were found in women when comparing to men in the following dimensions: “*Caring by connecting with others*” (*p* = 0.012) and “*Working with interpersonal and group differences*” (*p* = 0.028). Additionally, women obtained significantly higher scores than men in the following items of the ESSQ: item 10 “*I tolerate different ethical points of view in my surroundings*” (*p* = 0.028), item 11 “*I think it is good that my closest friends think in different ways*” (*p* = 0.005), item 12 “*I get also long well with those people who not agreeing with me*” (*p* = 0.010), item 13 “*I take other peoples’ points of view into account before making any important decisions in my life*” (*p* = 0.003), and item 16 “*I try to consider other peoples’ needs even in situations concerning my own benefits*” (*p* = 0.016).

### 3.2. Moral Sensitivity

Table 3 shows the results of the RMSQ. The RMSQ total score was 40.58 (5.36) points over 54. Regarding the dimensions of the questionnaire, the maximum score was obtained in the dimension “Moral strength” (4.97 (0.84) over 6), while the minimum score was obtained in the dimension “Sense of moral burden” (3.96 (0.90) over 6). When analyzing item by item, the maximum score was obtained in item 1 “*I always feel the responsibility to ensure that patients receive care, even if the resources are insufficient*” (5.23 (1.14) points over 6) and the minimum score was obtained in item 8 “*My ability to perceive the needs of the patient means that I frequently find myself in situations where I feel inadequate or uncomfortable*” (3.26 (1.38) points over 6).

When comparing moral sensitivity between women and men, we found no differences in RMSQ total score (39.95 (5.33) vs. 41.52 (5.35), respectively, *p* = 0.243) nor in any of the dimensions of the questionnaire. Only item 4 “*My ability to perceive the patient’s needs means that I do more than I have the strength to do*” showed significant differences when comparing women and men (3.67 (1.52) vs. 4.44 (1.12), respectively, *p* = 0.027).

## 4. Discussion

To the best of our knowledge, this is the first study that analyzes ethical sensitivity and moral sensitivity in practicing physical therapists. Scientific evidence about ethical sensitivity and moral sensitivity in this healthcare profession is scarce. Our results show that physical therapists reported high levels of ethical and moral sensitivity, and that women showed higher ethical sensitivity when compared to men, whilst no differences by gender were obtained in moral sensitivity.

In relation to ethical sensitivity, the ESSQ total score of our sample was 116.14 (5.87) over 140; thus, high levels of ethical sensitivity were shown, while women showed higher ethical sensitivity than men. Studies reported in other health professionals, such as nurses or physicians, reported moderate levels of ethical sensitivity [29,30]. With regards to nurses, Mert et al. [29] reported an ESSQ total score of 89.00. Similarly, Citlik et al. [30] reported an ESSQ total score of 91.5 (21.66), which is lower than the scores obtained in our study. Regarding the assessment of ethical sensitivity in physicians, Öztürk et al. [31] reported a ESSQ total score of 89.70 (17.00).

One of the possible determinants in the acquisition of ethical sensitivity skills was stress when performing daily practice. Considering that nurses and physicians reported high stress levels [32,33,34,35] this fact could justify the differences reported in ESSQ total scores when comparing to our sample of physical therapists. Another conditioning factor reported by previous studies that may influence ethical sensitivity is the level of knowledge of the skills related to clinical practice [29]. In this regard, it seems that education could influence the ethical sensitivity, since in our study half of the participants held a postgraduation, a master certification or a PhD dissertation; thus, these results are in line with another study performed in physicians [31].

In our study, women maintained a more holistic and integral view of health and considered other aspects of patient care beyond technical issues, these results being similar to those carried out for physicians [31,32,33]. Galiano Coronil et al. [36] found that women reported a more holistic care of patients than men (75.3 vs. 24.7%, respectively). By contrast, Bolanderas suggested no differences between male and female healthcare professionals when providing scientific biomedical information to patients [37]. In our study we observed differences in terms of healthcare professionals by gender and emphasized several aspects, such as the ability to communicate with patients, thus we agree with Galiano-Coronil et al. [36], who observed that, when compared to men, women exhibited greater sensitivity (41.1% vs. 24.9%, respectively) to social groups with health problems or difficulties in accessing healthcare, as well as greater interest in prevention and healthcare education (53.2% vs. 35%, respectively). This situation may be due to differences in cross-cultural sensitivity based on gender. Cultural awareness could influence the therapist’s ability to manage societal biases and to respect diversity [37,38]. In this sense, it seems that the level of awareness is considered a developmental process that evolves over time [19] and may affect daily clinical practice when physical therapists apply treatments to their patients. In addition, biases and inequalities with regards to gender seem to still exist in healthcare [36,39,40]. It seems that there is a tendency for men to be less socially sensitive, while women tend to be more sensitive to interpersonal needs and are the bearers of the emotions of culture [41].

With regards to moral sensitivity, the RMSQ total score of our study was 40.58 ± 5.36 over 54, therefore the sample showed high levels of moral sensitivity. By contrast, Nora et al. [42] performed a study with 100 nurses from Brazil who completed the 28-item modified Moral Sensitivity Questionnaire and reported nurses’ moral sensitivity to be moderate. Although this finding was congruent with our study, it should be taken into account that a different questionnaire was used to measure this outcome. Basar and Cilingir [26] found moderate levels of moral sensitivity in nurses working in intensive care units in Turkey, as well as differences in moral sensitivity according to the number of years of nursing experience, and to the duration of work as an intensive care nurse). Moreover, similar studies conducted in pediatric nurses [43], clinical nurses [44], and critical care nurses [45], reported moderate levels of moral sensitivity, contrary to our study. This variation may be due to different social and cultural contexts more than differences between physical therapists’ and nurses’ moral sensitivity.

When comparing moral sensitivity by gender in our study, no differences were observed. This finding is in line with another study carried out by Alyousei et al. [23], who did not find any difference by gender in nurses´ moral sensitivity (91.1 vs. 89.2 scores in women and men, respectively). Carmona and Montalvo [46] also did not obtain significant differences in terms of gender in moral sensitivity when assessing nurses (women scored 92.8 (8.3) and men scored 91.4 (11.1)). Similar results were observed in the study performed by Lee et al. [47]. Nevertheless, Lützén et al. [23] observed that women reported higher scores than men only in one item (“*When there are different views on goals, it is first of all the patient’s wishes that should be considered*”) and did not find any other differences by gender in their study. Finally, few studies have found differences in the moral sensitivity of healthcare professionals when comparing by age, marital status, education or years of experience [23,26,48,49].

Considering that quality of care may be influenced by the ethical awareness or even by the ethical and moral sensitivity that healthcare professionals exhibit, it seems to be important to assess these outcomes and to promote ethics-based courses to improve daily clinical practice. Therefore, understanding practicing physical therapists´ characteristics in relation to ethical sensitivity and moral sensitivity seems to be essential in this regard. In addition, the study and assessment of ethical and moral sensitivity should not only involve physical therapists, but the rest of healthcare professionals who care for patients daily in healthcare settings, in order to improve quality of care. This study presents a number of limitations that should be taken into account in future research. First, this was a cross-sectional study, and a larger sample size should be assessed. Second, our sample consisted of more women than men and was restricted to a specific geographic location. Third, we used a quantitative design to evaluate ethical and moral sensitivity by self-reporting and this approach could lead to reporting bias. Fourth, this study used self-reported questionnaires, thus it should be taken into account that only the participants´ self-perceived sensitivity was assessed, which may differ from their true ability or from what is actually practiced in stressful working conditions. As such, any generalization from the results should be made cautiously.

Despite the limitations, ethical and moral sensitivity in physical therapy professionals is not commonly studied despite its importance; in fact, to date, this is the first study that explores such outcomes in practicing physical therapists.

We further highlight that the assessment of moral and ethical sensitivity may help to design programs directed to improve the care of patients in their professional practice as healthcare professionals. Finally, future studies should be performed with greater samples, equal gender participation, in different geographic locations with multi-centric designs, and may use mixed-methods designs including semi-structured interviews.

## 5. Conclusions

Physical therapists showed high levels of ethical sensitivity and moral sensitivity. Women reported higher ethical sensitivity than men and obtained higher scores in caring by connecting with others, and in working with interpersonal and group differences. However, no differences by gender were obtained in moral sensitivity.

Our study highlights some interesting findings which may help to promote the design and implementation of ethics-based programs in physical therapists. Physical therapists are unique in their approach and in their ability to care for patients, and it should be taken into account that the quality of care may be influenced by the ethical and moral sensitivity that physical therapists exhibit. Understanding physical therapists´ characteristics in relation to ethical and moral sensitivity is essential to design and implement ethics-based courses directed to improve the care of patients in their daily clinical practice.

## Figures and Tables

**Table 1 healthcare-11-00333-t001:** Sociodemographic characteristics of the sample.

	Total *n* = 75	Women *n* = 44	Men *n* = 31
Age, years, mean (SD)	34.56 (8.68)	33.21 (7.41)	36.64 (10.13)
Gender, frequency (%)		44 (58.70)	31 (41.30)
Education, frequency (%)			
Diploma in Physical Therapy	24 (32.00)	13 (29.50)	11 (35.50)
Degree in Physical Therapy	9 (12.00)	4 (9.10)	5 (16.10)
Postgraduation in Physical Therapy	6 (8.00)	3 (6.80)	3 (9.70)
Master in Physical Therapy	28 (37.30)	19 (43.20)	9 (29.00)
Doctoral thesis in Physical Therapy	8 (10.70)	5 (11.40)	3 (9.70)
Previously completed an ethics course, frequency (%)			
Yes	8 (10.70)	2 (4.50)	6 (19.40)
No	67 (89.30)	42 (95.50)	25 (80.60)
Years of experience as a physical therapist, mean (SD)	12.23 (8.16)	11.75 (7.49)	12.90 (9.16)

All data are presented as mean (SD) or as frequency (%), as appropriate. %: percentage; SD: Standard deviation.

**Table 2 healthcare-11-00333-t002:** Results of the Ethical Sensitivity Scale Questionnaire.

Ethical Sensitivity Scale Questionnaire	Total *n* = 75	Women *n* = 44	Men *n* = 31	*p*-Value between Groups
Dimension 1. Reading and expressing emotions	4.02 (0.71)	4.05 (0.71)	3.97 (0.73)	0.680
Dimension 2. Taking the perspectives of others	4.61 (0.64)	4.69 (0.71)	4.49 (0.52)	0.183
Dimension 3. Caring by connecting with others	4.27 (0.67)	4.43 (0.70)	4.04 (0.56)	**0.012**
Dimension 4. Working with interpersonal and group differences	4.24 (0.75)	4.40 (0.71)	4.01 (0.77)	**0.028**
Dimension 5. Preventing social bias	3.69 (0.63)	3.76 (0.65)	3.59 (0.60)	0.264
Dimension 6. Generating interpretations and options	4.26 (0.69)	4.38 (0.67)	4.10 (0.69)	0.084
Dimension 7. Identifying the consequences of actions and options	3.83 (0.71)	3.94 (0.69)	3.69 (0.71)	0.130
Total score	116.14 (15.87)	119.21 (16.26)	111.43 (14.28)	**0.043**

All data are presented as mean (SD). SD: Standard deviation. Significant differences are highlighted in bold.

**Table 3 healthcare-11-00333-t003:** Results of the Revised-Moral Sensitivity Questionnaire.

Revised-Moral Sensitivity Questionnaire	Total *n* = 75	Women *n* = 44	Men *n* = 31	*p*-Value between Groups
Dimension 1. Sense of moral burden	3.96 (0.90)	3.87 (0.92)	4.10 (0.87)	0.270
Dimension 2. Moral strength	4.97 (0.84)	4.95 (0.84)	5.00 (0.86)	0.820
Dimension 3. Moral responsibility	4.57 (1.05)	4.61 (1.04)	4.52 (1.06)	0.694
Total score	40.58 (5.36)	39.95 (5.33)	41.52 (5.35)	0.243

All data are presented as mean (SD). SD: Standard deviation.

## Data Availability

Not applicable.

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
