# Peer review of "Physical Therapists’ Ethical and Moral Sensitivity: A STROBE-Compliant Cross-Sectional Study with a Special Focus on Gender Differences"

_healthcare, 2023, doi:10.3390/healthcare11030333_

Round 1

Reviewer 1 Report

Reviewer Report:   In this work, the authors aimed to explore ethical sensitivity and moral sensitivity in practicing physical therapists, and to compare both variables by gender.   They asked 75 physical therapists (58.7% women; average age=34.56(8.68) years) to fill in questionnaires measuring ethical sensitivity (Ethical Sensitivity Scale Questionnaire) and moral sensitivity (Revised-Moral Sensitivity Questionnaire).   The sample indicated high ethical sensitivity (116.14±15.87 over 140) and high moral sensitivity (40.58±5.36 over 54).   They comparing by gender, women reported significantly higher ethical sensitivity than men (p=0.043), as well as higher scores in the following dimensions: Caring by relating with others (p=0.012) and working with interpersonal and group differences (p=0.028). But, However, no differences were found in moral sensitivity (p=0.243).   Finally, they concluded that physical therapists indicated high levels of ethical and moral sensitivity, whilst women reported higher ethical sensitivity than men. Understanding physical therapists´ ethical and moral sensitivity is essential to design and implement integrated education programs directed to improve the quality of care of patients in their daily clinical practice.       

The cited references are relevant to the research.

The research design is appropriate.

The results are presented, clearly.

The conclusions are supported by the results.

After controlling the attached paper with minor revision, I recommend that the paper can be publish in the Healthcare journal.

Author Response

Comments for the Editors and Reviewers: 

Response to reviewer

Thank you very much for your highly relevant comments to our manuscript. Please find below our point-by-point response to the reviewer comments which we hope will meet the requirements for acceptance. All suggestions have been implemented and discussed. All revisions have been formatted with track-changes in the revised manuscript. 

Reviewer 1. In this work, the authors aimed to explore ethical sensitivity and moral sensitivity in practicing physical therapists, and to compare both variables by gender. They asked 75 physical therapists (58.7% women; average age=34.56(8.68) years) to fill in questionnaires measuring ethical sensitivity (Ethical Sensitivity Scale Questionnaire) and moral sensitivity (Revised-Moral Sensitivity Questionnaire). The sample indicated high ethical sensitivity (116.14±15.87 over 140) and high moral sensitivity (40.58±5.36 over 54). They comparing by gender, women reported significantly higher ethical sensitivity than men (p=0.043), as well as higher scores in the following dimensions: Caring by relating with others (p=0.012) and working with interpersonal and group differences (p=0.028). But, However, no differences were found in moral sensitivity (p=0.243).   Finally, they concluded that physical therapists indicated high levels of ethical and moral sensitivity, whilst women reported higher ethical sensitivity than men. Understanding physical therapists´ ethical and moral sensitivity is essential to design and implement integrated education programs directed to improve the quality of care of patients in their daily clinical practice.      

The cited references are relevant to the research. The research design is appropriate. The results are presented, clearly. The conclusions are supported by the results. After controlling the attached paper with minor revision, I recommend that the paper can be publish in the Healthcare journal.  

Answer: Thank you for your comments.

Reviewer 2 Report

Dear Authors

Thank you for the opportunity to review this manuscript.

This is a cross sectional study. The authors have explored the aimed to explore ethical sensitivity and moral sensitivity in practicing physical therapists, and to compare both variables by gender. 

They conclude that Physical therapists showed high levels of ethical and moral sensitivity, with a difference between women respect to men.

I think that this paper provides a useful addition to the literature. It is generally well presented, concise and clear, despite the small sample size and the heterogeneity of population.

I suggest minor revision:

- a general check of English is needed

- There are grammatical mistakes to be correct (eg in line 43: atenttion)

- Discussion should be improved

Author Response

Comments for the Editors and Reviewers: 

Response to reviewer

Thank you very much for your highly relevant comments to our manuscript. Please find below our point-by-point response to the reviewer comments which we hope will meet the requirements for acceptance. All suggestions have been implemented and discussed. All revisions have been formatted with track-changes in the revised manuscript. 

Reviewer 2:

 Dear Authors: Thank you for the opportunity to review this manuscript. This is a cross sectional study. The authors have explored the aimed to explore ethical sensitivity and moral sensitivity in practicing physical therapists, and to compare both variables by gender. They conclude that Physical therapists showed high levels of ethical and moral sensitivity, with a difference between women respect to men. I think that this paper provides a useful addition to the literature. It is generally well presented, concise and clear, despite the small sample size and the heterogeneity of population. I suggest minor revision:

- a general check of English is needed

Answer: The authors thank the reviewer for his/her appreciation. Language revision has been performed.

- There are grammatical mistakes to be correct (ex in line 43: attention)

Answer: Thank you for the comment, we have reviewed the whole manuscript and corrected the typos and mistakes.

- Discussion should be improved

Answer: Thank you for your comment. We have proceeded to perform the changes and corrections in the discussion, as well as added new references to this section.

Reviewer 3 Report

This study in presents self reported data on ethical and moral sensitivity amongst physiotherapists. It is an interesting topic as ethical and moral sensitivity is not commonly studied.

A few major concerns

1. Its not clear from the introduction what is the difference between ethical and moral sensitivity. The elaborations in lines 60-80 are vague and although lengthy, does not provide clear definitions or understanding. Much of the definition is circular e.g. ethical sensitivity is the ability to detect ethical problems. It would help to try to define "ethical sensitivity" without repeating the word ethical, and it would be important to define what is 'ethical', rather than defining just the 'sensitivity' aspect. 

Similarly, avoid using the term moral to define moral sensitivity. 

The use of illustrations and examples might help with defining these terms. It may also help to list/tie in some of the domains assessed in the respective questionnaire in this section, so that the definition is a bit more concrete.

2. I'm worried about the presence of social desirability/reporter bias. Its not clear, but this seems like a single centre study. The questionnaires were also administered face to face. It is quite likely that participants will feel some pressure (whether subconscious or conscious) to provide the 'correct/ideal' answer. This is especially problematic since the 'ideal' answer is obvious from the questions.

Furthermore, even if participants were wholeheartedly truthful, these questionnaires only assess the participants self-perceived sensitivity, which may differ drastically from their true ability or what is actually practised in stressful working conditions. This limitation should be acknowledged.

3. The conclusion of high ethical and moral sensitivity in this cohort cannot be made without some form of comparison with either other healthcare providers or other cohorts. For all we know, the mean scores in physiotherapists are below average.

4. It is currently hard to appreciate clinical utility or implications of these results. It does not provide evidence to suggests that these things are important to measure, nor that they change outcomes. Author may consider pivoting to discussing additional studies (with more clinical value) that can build upon this study's results; for example what settings or contexts where ethical/moral sensitivity is likely compromised, does training improve this, does this impact on patient outcomes or physicial wellbeing/burnout etc

Author Response

Comments for the Editors and Reviewers: 

Response to reviewer

Thank you very much for your highly relevant comments to our manuscript. Please find below our point-by-point response to the reviewer comments which we hope will meet the requirements for acceptance. All suggestions have been implemented and discussed. All revisions have been formatted with track-changes in the revised manuscript. 

Reviewer 3. This study in presents self reported data on ethical and moral sensitivity amongst physiotherapists. It is an interesting topic as ethical and moral sensitivity is not commonly studied. A few major concerns:

  1. Its not clear from the introduction what is the difference between ethical and moral sensitivity. The elaborations in lines 60-80 are vague and although lengthy, does not provide clear definitions or understanding. Much of the definition is circular e.g. ethical sensitivity is the ability to detect ethical problems. It would help to try to define "ethical sensitivity" without repeating the word ethical, and it would be important to define what is 'ethical', rather than defining just the 'sensitivity' aspect. The use of illustrations and examples might help with defining these terms. It may also help to list/tie in some of the domains assessed in the respective questionnaire in this section, so that the definition is a bit more concrete.

Answer: Thank you for your pertinent comment. The definitions of ethical sensitivity and moral sensitivity have been revised and the writing has been improved in this regard. We have made changes and added references to clarify this matter, avoiding circular definitions. Also, we have differentiated the nuances that characterize each term and also listed the domains of ethical and moral sensitivity in the introduction section.

  1. I'm worried about the presence of social desirability/reporter bias. It’s not clear, but this seems like a single center study. The questionnaires were also administered face to face. It is quite likely that participants will feel some pressure (whether subconscious or conscious) to provide the 'correct/ideal' answer. This is especially problematic since the 'ideal' answer is obvious from the questions. Furthermore, even if participants were wholeheartedly truthful, these questionnaires only assess the participants self-perceived sensitivity, which may differ drastically from their true ability or what is actually practiced in stressful working conditions. This limitation should be acknowledged.

Answer: Thank you for your interesting comments. Although the contact was face-to-face, once the informed consent was signed and sociodemographic data were collected, the researcher provided the questionnaires to each participant and privacy was ensured for the participant in order to complete the self-reported questionnaires. The researcher was not present when participants gave their responses to the questionnaires. In order to clarify this information, we proceeded to improve the explanation in methods. Finally, regarding the use of self-reported questionnaires, we agree with the reviewer and therefore we have added a new limitation in this regard, including the information suggested by the reviewer.

  1. The conclusion of high ethical and moral sensitivity in this cohort cannot be made without some form of comparison with either other healthcare providers or other cohorts. For all we know, the mean scores in physiotherapists are below average.

Answer: Thank you for your valuable comment. The results of our study indicate that the sample of physical therapists showed high levels of ethical and moral sensitivity, as absolute values, based on the total score obtained in each questionnaire over the maximum score of each questionnaire. However, we have proceeded to improve the discussion and added several references related to other studies conducted in healthcare professionals, in order to compare the ethical and moral sensitivity, and concretely the total score of the ESSQ and RMSQ questionnaires. In addition, we have also added new references regarding gender differences when assessing ethical and moral sensitivity in healthcare professionals.

Round 2

Reviewer 3 Report

The previous comments have been appropriately addressed, i have no further to add